# A Review of Polyurethane Foams for Multi-Functional and High-Performance Applications

**DOI:** 10.3390/polym16223182

**Published:** 2024-11-15

**Authors:** Huanhuan Dong, Shujing Li, Zhixin Jia, Yuanfang Luo, Yongjun Chen, Jiang Jiang, Sheng Ji

**Affiliations:** 1Key Lab of Guangdong High Property and Functional Macromolecular Materials, School of Materials Science and Engineering, South China University of Technology, Guangzhou 510640, China; hhdong536@163.com (H.D.); lee0_cn@163.com (S.L.);; 2Justape New Material Technology Co., Ltd., Heyuan 517135, China

**Keywords:** polyurethane foams, flame retardancy, electromagnetic interference shielding, sound absorption, sustainability

## Abstract

Polyurethane (PU) foams are cellular polymeric materials that have attracted much attention across various industries because of their versatile properties and potential for multifunctional applications. PU foams are involved in many innovations, especially in multi-functional and high-performance applications. Special attention is given to developing tailored PU foams for specific application needs. These foams have various applications including flame retardancy, sound absorption, radar absorption, EMI shielding, shape memory, and biomedical applications. The increasing demand for materials that can perform multiple functions while maintaining or enhancing their core properties has made PU foams a focal point of interest for engineers and researchers. This paper examines the challenges faced by the PU foam industry, particularly in developing multifunctional products, as well as the strategies for improving sustainability, such as producing PU foams from renewable resources and recycling existing materials.

## 1. Introduction

Polyurethane (PU) foams have emerged as versatile and indispensable materials across a wide range of multifunctional and high-performance applications. PU foams have low density and good dimensional stability due to their hollow cellular structure and chemical composition. PU foams are renowned for their unique combination of properties, including elasticity, low density, and good air permeability. The performance of PU is primarily determined by its molecular structure and the aggregation state, which consists of soft and hard phases. The soft phases should be in proper ratio with the hard phases for the optimum performance of PU. For multifunctional uses, PU foams can be tailored to specific requirements by adjusting their chemical composition conditions. The flexibility enables the creation of foams with varying densities, hardness, flame retardancy, sound absorption, radar absorption, and electromagnetic interference shielding. As a result, PU foams have many industrial and everyday applications, such as thermal insulation in buildings, sound absorption, radar absorption, electromagnetic interference shielding, shape memory, and biomedical materials, all of which exhibit promising development prospects [1].

PU foams are composed of soft and hard segments with a micro-phase separation structure, where the type and content of the hard segments are key factors influencing their mechanical properties and heat resistance [2]. The performance of PU is primarily determined by its molecular structure and the aggregation state, which consists of soft and hard phases. Currently, the soft phases of PU are typically polyol. The hard phase is more diverse, primarily formed by the reaction between diisocyanate, chain extenders, and the terminal hydroxyl groups of the soft phase. PU foam formation is a complex polymerization process involving at least seven types of primary and auxiliary materials. This process is highly sensitive to material temperature and the timing of reactions, which usually occur within 1~2 min, requiring precise production techniques [3].

PU foams have garnered significant research attention due to their versatility. A variety of foaming techniques have been employed to produce PU foams [4]. Rigid PU foams possess valuable properties, including high mechanical strength, excellent weather resistance, low thermal conductivity, low density, and superior damping characteristics. These properties are ideal for advanced applications. Flexible PU foams have been developed with useful features. A common method to enhance foam properties is by reinforcing them with various nanofillers [5]. Fillers are typically incorporated into polyol, and prepolymer in PU synthesis [6]. The combination of foaming technology and nanoparticles promotes the development of multifunctional PU composite foams [7].

For high-performance applications, advanced PU foams with superior mechanical properties are being developed. PU foams are capable of withstanding heavy loads and are used in industrial applications, such as the production of composite materials and structural components. Additionally, PU foams are tailored to specific requirements by adjusting their chemical composition conditions [8]. The flexibility enables the creation of foams with varying densities, hardness, flame retardancy, sound absorption, radar absorption, and electromagnetic interference shielding. PU foams hold great promise for multi-functional and high-performance applications. However, challenges remain, particularly regarding the recycling and disposal of PU foams, which pose environmental concerns [9]. Ongoing research is focused on identifying sustainable solutions for the end-of-life management of these materials [10].

## 2. PU Foams

Polyurethane (PU) foams are a polymeric material with a cellular structure, widely used in various applications, from household items to transportation. PU foams demonstrate good properties and characteristics. Flexible PU foams, characterized by their open-cell structure, are particularly prone to flaming and release highly toxic and combustible gasses during combustion [11]. Significant research has been conducted to enhance the pyrolysis behavior, flammability, and other properties of PU foams.

PU foams are recognized as excellent materials for various applications due to their low density, good dimensional stability, and so on. They are suitable for use in numerous industrial applications. The performance of PU foams depends on the raw materials and processing techniques. Numerous nanomaterials, such as carbon nanotubes, graphene derivatives, clay, and silica, have been added to PU foams to enhance their mechanical properties, thermal stability, biodegradability, and so on. This characteristic of PU foams allows for their adaptation to meet specific needs by modifying their chemical composition and processing conditions, resulting in foams with various densities, hardness, flame retardancy, sound absorption, radar absorption, and electromagnetic interference-shielding properties.

### 2.1. Flame Retardant

PU foams are prone to decomposition and combustion when exposed to high temperatures [12]. During combustion, they release highly toxic gasses, including carbon monoxide and hydrogen cyanide, which pose serious health risks and can be fatal upon inhalation. Additionally, the rise in temperature during a fire accelerates the decomposition of PU foams and, when mixed with air, creates a flammable mixture. Combustion also produces melting drips and large amounts of toxic fumes, making PU foams unsuitable for many industrial applications [13].

Developing flame-retardant PU foams has thus become a critical area of research. The flame resistance of PU foams can be improved by introducing flame-retardant agents (chlorine, bromine, and phosphorus compounds) during the foaming process. Traditionally, halogen-based flame retardants were used, but their toxicity and environmental impact have driven the search for alternatives. Jiang et al. developed alginate-based PU foams with good flame-retardant properties using an in situ chemical foaming method [14]. However, the combustion of these materials still produces significant amounts of corrosive gasses, harmful to both the environment and human health [15].

Surface modification techniques have gained attention to enhance the flame-retardant properties of PU foams. Li et al. developed flame-retardant PU foams by coating them with polydopamine and treating them with hexamethyldisilane [16]. Similarly, Ma et al. prepared an interpenetrating polymer network (IPN) between PU foams and porous polymers, including phosphorus and nitrogen elements, improving both flame retardancy and mechanical strength [17]. Jamsaz et al. produced graphene-based flame-retardant PU foams by functionalizing them with graphene oxide [18]. Pan et al. [19] created a multilayered lanthanum phenylphosphonate (LaPP) film on the PU foams as shown in Figure 1. The treated foams exhibited flame-retardant properties.

Traditional halogen-based flame retardants need to be replaced due to their toxicity and environmental impact. The surface modification process and the development of new organic flame-retardant components have become two popular methods, that is, the use of some flame-retardant components on the surface of the foam for modification, or the modification of the basic components of the foam. Commonly used materials include modified polyols containing phosphorus and nitrogen, graphene oxide, phosphoric acid, etc. To reduce the smoke generation of polyurethane foam when burned, inorganic flame retardants including magnesium hydroxide, titanium dioxide, and silicon dioxide have been explored. Expandable graphite (EG) can reduce the effective heat of combustion (EHC) by inhibiting heat release during matrix decomposition into gas products, indicating its significant role in enhancing fire resistance in the condensed phase. Zhang et al. [20] used castor oil-based polyols to prepare PU foams with good flame retardant. With the higher flame-retardant polyol content, the mechanical properties of the PU foams increased. The PU foams demonstrated good fire resistance. Zhang et al. [21] prepared flame-retardant PU foams by adding castor oil phosphate flame-retardant polyol (COFPL) and expandable graphite (EG) (Figure 2). The PU foams show good flame retardancy properties due to the addition of COFPL.

To address the issues of poor compatibility, Rao et al. [22] prepared a polyester polyol, and used polyester polyol to prepare PU foams with good flame-retardancy properties. This was achieved via ester-exchange reactions using dimethyl methylphosphonate and diethanolamine as raw materials. Flexible PU foams with 10 phr synthesized polyester polyol have good fire resistance. Lubczak et al. [23] prepared boron-containing PU foams with good flame-retardancy properties by adding oligoetherol. The foams exhibited a high oxygen index.

While organic modifications of PU foams can enhance flame-retardant performance, they often generate significant smoke during combustion and exhibit suboptimal flame-retardant properties [24]. To mitigate these issues, inorganic flame retardants, such as magnesium hydroxide [25], titanium dioxide [26], and silicon dioxide [27], have been explored. Wang et al. prepared PU foams with SiO_2_/graphene oxide. The PU foams exhibited a 45% reduction in peak heat release rate (pHRR) [28], alongside decreased hydrogen chloride gas production, lower toxic emissions, and excellent thermal stability at high temperatures. However, the process remains complex and expensive.

PU foams have also been modified with graphene oxide, polystyrene sulfonate, and SiO_2_ through a soaking method. The method aims to form a stable structure and an inorganic SiO_2_ on the surface of PU foams. The resulting PU foams have excellent flame-retardant performance (Figure 3) [29]. Xu et al. [30] prepared PU foams by incorporating montmorillonite alongside phosphorus-based flame retardants.

Expandable graphite (EG) can also decrease the effective heat of combustion (EHC) by inhibiting heat release during matrix decomposition into gas products. Chen et al. [31] prepared a flame-retardant PU foam by introducing 3-(N-diphenyl phosphate) and modified EG with phosphorus-containing ionic liquid. The PU foam exhibits excellent thermal stability and carbon production. Chen et al. [32] prepared rigid PU foams with EG composites via the liquid blending method (as illustrated in Figure 4). Their investigation revealed that a 15 phr EG loading produced optimal flame retardancy. EG decomposes before PU foams under thermal treatment, forming a dense, carbonized layer on the surface of the foams. The layer impedes the transfer of oxygen and heat into the foams, significantly improving flame-retardancy properties.

The higher the EHC, the greater the hazardousness of material when burned. The inclusion of EG helps reduce this parameter, indicating its significant role in enhancing fire resistance in the condensed phase. Like most organic materials, PU foams are highly flammable. Nevertheless, for many years, their fire performance has been considered acceptable. However, as fire safety standards become increasingly stringent, there is a growing need to enhance the flame-retardancy properties to meet future regulatory requirements for PU foams.

### 2.2. Electromagnetic Interference Shielding

Wave-absorbing materials can achieve stealth and counter-stealth capabilities in electronic devices. PU foam-based wave-absorbing composite materials exhibit excellent electromagnetic absorption properties and strong adaptability in microwave absorption, making them highly versatile. These materials hold significant potential for future development. A schematic of the electromagnetic interference (EMI) shielding mechanism in PU foams is illustrated in Figure 5 [33].

PU composite foams are suitable for use in the field of absorbing shielding due to their lightweight, high processability, and insensitivity to corrosive environments. The electromagnetic interference (EMI) shielding effectiveness value of PU foam composites can reach 50~60 dB, far surpassing the target level of ~20 dB required for commercial application. To enhance the conductivity of PU foams, various conductive fillers such as carbon black, graphene, and CNTs have been introduced into the PU matrix. Rigid wave-absorbing PU foams are synthesized by incorporating wave-absorbing agents into a rigid PU foam system through a mold foaming reaction. Due to the controllability of density, strength, and hardness through raw material formulation, as well as the ease of shaping and processing, these materials are widely utilized in various stealth systems for weapons. However, when used alone, rigid PU foams can be brittle and exhibit poor stiffness, making them susceptible to fracture, particularly in larger sizes.

Flexible PU foams are shaped by the reactive foaming of raw materials, followed by impregnation with an absorbent and subsequent drying to produce the final product. Wave-absorbing PU foams are primarily employed in non-echo darkrooms due to their propensity for water absorption, low processing precision, and limited weather resistance. While flexible foam wave-absorbing materials exhibit poorer environmental adaptability compared to rigid foams, they offer favorable broadband characteristics, lightweight properties, good flexibility, and excellent vibration resistance, thus holding significant potential for application despite the risk of breakage.

Nowadays, there is an increasing demand for electrical conductivity. The rising interest in electronics (such as communication, computation, and automation) has led to greater EMI, resulting in environmental pollution. Conductive PU foam composites have emerged as effective candidates for EMI shielding [33]. To enhance these properties, various fillers such as carbon black, graphene, and CNTs can be employed [34]. Compared to zero-dimensional carbon black and three-dimensional graphite, one-dimensional carbon fibers and carbon nanotubes, as well as two-dimensional graphene, possess larger aspect ratios or diameter-to-thickness ratios. Consequently, PU foams formulated with these materials typically demonstrate higher conductivity at similar filler contents. Figure 6 shows carbon-based fillers for EMI shielding effectiveness.

Kim et al. [35] reported the electromagnetic interference (EMI) shielding of PU foams containing carbon fiber. The study reported that the composite exhibited improved EMI shielding performance. CNTs are known for their exceptional electrical properties. CNTs are promising candidates for enhancing the electrical performance of PU foams. They [35] examined the impact of CNTs on the EMI shielding performance of PU foams, finding that longer nanotubes yield better performance.

Graphene and graphene oxide (GO) are considered effective additives for increasing the EMI shielding performance of PU foams. Jeddi et al. [36] reported that good EMI shielding performance arises from the three-dimensional graphene networks in the PU foams. Ahmad et al. [37] prepared absorber PU foams by adding graphite nanosheets (GNs) and silicone rubber. The PU foams have good EMI shielding performance due to the GNs network. Furthermore, the rise in conductivity of the coarse nanocomposite accelerates energy storage in the network during each oscillation period, resulting in higher real permittivity.

### 2.3. Sound Absorption and Damping PU Foams

Another application where PU foams excels is sound absorption. Noise pollution poses significant challenges in the automotive industry [38]. The major mechanisms for sound wave absorption involve friction with damping by the absorbing PU foams [39]. A schematic representation of the sound absorption for PU foams is shown in Figure 7 [40], alongside a diagram illustrating the mechanisms of sound absorption of PU foams.

In recent material developments, sound absorption through damping has emerged as a critical factor in sound absorption by converting sound energy into thermal energy via hysteresis. Numerous modeling studies have demonstrated a significant relationship between these non-acoustic parameters and the microcellular structure of PU foams. The sound absorption efficiency could be enhanced by changing the cellular structure of PU foams. The cellular structure of PU foams is directly affected by the primary ingredients used in their fabrication, such as polyols, isocyanates, and so on [41].

Gwon et al. [39] investigated the impact of cellular structure on the sound absorption behavior of PU foams. Different gelling catalysts and varying water contents were employed to create pore structures, which are presented in Figure 8. The catalyst in the urethane reaction resulted in many small pores compared to the less active catalyst. A higher concentration of small cavities was achieved at elevated water content. While sound absorption efficiency is generally low at lower foam densities, the findings indicate that the reduction in foam density results in an increase in sound absorption efficiency.

The molecular structure of PU foams significantly influences the formation of interconnecting pores. The incorporation of modified isocyanate-containing uretonimine linkages facilitates the microphase separation within the PU foams. Furthermore, decreasing the content of toluene diisocyanate promotes phase separation among the hard phases of PU. The sound absorption capability of the foams was increased [42].

The addition of nanofillers can increase the number of open structures on the foam wall, or forms a complex channel structure, which is conducive to an improvement in the sound absorption performance of the foams. Sound-damping PU foams are produced by adding nanofillers and foaming agents, with sonication commonly employed during the mixing process to prevent filler aggregation [43]. Figure 9 illustrates the SEM images and sound-damping performance of PU foams, foams without sonication, and hybrid foams with sonication. The hole size of the foam significantly reduced following the sonication treatment, and the sonication treatment improved the sound absorption capability of the PU foams. And functionalizing the foams through post-treatment enhancement can also improve sound-damping properties. Graphene oxide (GO) was utilized to modify PU foams to optimize airflow resistance. Typically, PU foams are soaked in a GO aqueous solution (Figure 9e) and subsequently heat-treated (Figure 9f). The PU foams demonstrate increasingly complex channels as density increases. In Figure 9j, PU foams with higher density exhibit a greater absorption coefficient at low frequencies, outperforming PU foams in terms of absorption capability. In addition to excellent sound absorption properties, PU foams with GO also possess moisture insulation and fire-retardant characteristics [44].

To enhance the sound absorption capability of PU foams with an open cell structure, plate-like fillers, such as bentonite, clay, graphene oxide, and montmorillonite intercalated with poly (ethylene glycol), have been added to PU foams. A kind of phosphorus-containing graphene oxide (D-GO) was designed, and the D-GO-filled PU foam was synthesized by employing vacuum impregnation technology. Compared to pure PU foam, the D-GO-filled PU foam shows enhanced smoke suppression, sound absorption, and noise reduction [38].

### 2.4. Other Properties and Applications of PU Foams

PU foams have emerged as significant polymers for biomedical applications because of their good properties, including biocompatibility, and controllable chemical and physical characteristics. These properties have proven effective for various biomedical purposes [45]. There are many applications of PU foams in biology, including drug delivery, tissue engineering, and long-lasting biomedical devices. The modification of PU foams has broadened their applicability in biomedical contexts, leading to their use in drug delivery, scaffolds, and stents for tissue, bio-fluid absorption, or biocatalytic air filters. Advancements in PU foams have led to innovative applications including drug delivery or tissue engineering [46]. Due to their stability and bio-sustainability, PU foams are preferred materials for long-lasting biomedical devices [47]. They exhibit excellent properties in injectable scaffolds. Schreader et al. [48] prepared PU-based foams containing hydroxyapatite nanoparticles, which demonstrated excellent biocompatibility. These PU foams can be used as bone tissue. Zawadzak et al. [49] created CNT-coated PU via electrophoretic deposition for tissue engineering. CNTs are nucleation centers, facilitating hydroxyapatite formation on foam surfaces compared to non-coated PU foams. CNT-coated PU foams are promising candidates for bioactive scaffolds for bone tissue applications due to their favorable porosity and bioactivity [50]. But using CNTs-based PU foams raises concerns regarding managing the levels of toxicity both in vivo and in vitro [51]. Further developments could result from future studies that concentrate on controlled toxicity levels.

Additionally, injectable delivery techniques have demonstrated the usefulness of PU foams and nanocomposite foams. In this context, PU/graphene and PU/CNTs foams scaffolds have been prepared. Shin et al. [52] developed 3D scaffolds based on graphene and graphene oxide nanosheet-filled PU foams, which were evaluated for their capacity to facilitate skeletal tissue cell growth. Additionally, PU nanocomposite foams have served as a scaffold for the regeneration of skeletal tissues, which have a pore size of approximately 300 μm. These PU foams create a favorable microenvironment for the growth of skeletal cells and myogenic skeletal cell differentiation. Studies indicate that PU/graphene and PU/graphene foams positively influence myogenic stimulation in myoblasts, making them effective in the design of 3D scaffolds for drug delivery or various applications. However, further research is needed to assess the potential harmful and long-term effects of nanocarbon-filled PU foams.

PU foams can also be utilized in other applications such as vascular grafts, heart valves, breast implants, and ophthalmic implants. However, for any material to be used in biomedical applications, excellent biocompatibility is essential. Conversely, the biostability of polylactic acid has raised concerns, as polyester-based polylactic acid is unstable in aqueous and oxygenated environments, and even polyether-based polylactic acid lacks stability. However, the biostability of PU has been enhanced through the incorporation of stable polysiloxanes. The advancements of PU foams have positioned them as an important biomedical material, leading to their use in drug delivery, scaffolds, and stents for tissue, bio-fluid absorption, or biocatalytic air filters.

## 3. Sustainable Development of PU Foams

PU foams have become essential materials in various industries due to their versatile properties, including comfort, insulation, sound absorption, and durability [53]. However, most polyols used to produce PU foams are made from raw materials derived from petroleum, and the management of increasing PU waste presents a significant challenge. The traditional production and disposal methods for PU foams have raised significant environmental sustainability concerns. The development of sustainable PU foams focuses on reducing environmental impact through the use of renewable resources, enhancing recyclability, and incorporating multifunctional properties [54].

The foundation of advancing sustainable PU foams lies in transitioning from fossil-derived raw materials to bio-based substitutes. Lignocellulosic biomass, an inexhaustible natural resource, has emerged as a leading candidate for supplying biopolyols, effectively replacing petroleum-based polyols in the composition of PU foams [54]. The structuring of lignocellulose biomass and the structural formula of the components, along with the applications of PU foams, are illustrated in Figure 10. This abundant vegetation is converted into valuable biopolyols through both acidic and alkaline catalytic breakdown mechanisms, thus reducing dependency on finite resources and ingeniously upcycling agricultural waste into high-value products. Furthermore, the integration of bio-derived flame-retardant agents enhances the safety characteristics of these environmentally friendly foams without compromising their green performance metrics.

Most polyols used to produce PU foams are made from raw materials derived from petroleum. However, growing concern about environmental problems and the depletion of petroleum have prompted the development of PU foams from biological and renewable feedstocks [55]. The use of sustainable products such as green and bio-based polyols in PU foams has increased as a result of this growing interest in bio-based materials. This shift is fueled by a preference for PU foams that have a lower carbon footprint and can be recycled or sourced from non-polluting sources. Consequently, bio-based polyols serve as viable alternatives to petroleum-based polyols. Furthermore, extensive research has focused on the development of renewable polyols sourced from biomass residues, vegetable, or industrial by-products [56].

Ji et al. [57] demonstrated that the incorporation of vegetable oil polyols into PU materials increased thermal stability. This enhancement is attributed to an increase in the crosslink density of PU foams, resulting in a more stable structure. Pawlik et al. [58] prepared flexible PU foams using homemade palm oil polyols. They found that the functionality of petroleum-based polyols (3.0) exceeded that of palm oil polyols (2.5). The addition of palm oil polyols improved the tensile strength of the PU foams because of differences in the composition and positioning of hydroxyl groups within the polymer backbone; specifically, the hydroxyl groups in palm oil polyols are in the middle of the chain, leading to a higher crosslinking density compared to petroleum-based polyols. Generally, the presence of long alkyl chains in palm oil polyol contributes to a greater number of soft segments, thereby enhancing elongation at break. The regularity in the structure of palm oil polyols and the positioning of hydroxyl groups result in a more homogeneous structure during the crosslinking reaction, leading to improved mechanical properties [59].

Campanella et al. [59] utilized methanol to open the ring of homemade epoxidized soybean oil to obtain polyol. Compared with hydroxylated and esterified soybean oils, the polyol produced through epoxidation exhibited the highest hydroxyl value and viscosity among the samples, making it most suited to the preparation of PU foams. Guo et al. [60] synthesized three different soybean oil polyols by epoxidizing soybean oil using alcohol, hydrochloric acid, and hydrobromic acid, respectively. Their results indicated that the polyols prepared with hydrobromic acid displayed the lowest hydroxyl value but the highest functionality and molecular weight. Additionally, the choice of ring-openers influenced the viscosity of the soybean oil polyols; among the three synthesized, only the polyol produced with methanol remained liquid at room temperature, while the others solidified.

As the production of PU foams continues to rise, the management of increasing PU waste presents a significant challenge. Addressing the end-of-life issues associated with PU foams is essential for achieving true sustainability [1]. Developing effective recycling methods and strategies for repurposing post-consumer PU foams is gaining traction. Chemical recycling methods that convert waste PU foams back into their constituent monomers or other valuable chemicals represent a promising avenue for sustainability [61].

To enhance the recyclable capabilities of PU foams and preserve their functionality and properties, Fortman et al. [62] reported on thermally recyclable PU incorporating Lewis acid catalysts. The catalysts promote the exchange of carbamate, which allows for the recycling of PU foams at high temperatures. The recycling of PU foams requires a catalyst to be added just before reprocessing. They used a method to recycle PU foams with an external catalyst. Their findings revealed that the recycled PU foams resulted in good mechanical properties.

Kim et al. [1] utilized a method of twin-screw extrusion to recycle PU foams, taking advantage of the melt processability of PU foams. Using two additives (zirconium (IV) and acetylacetonate [Zr(acac)4]) to facilitate the reforming of PU foams, they identify a replacement for toxic tin catalysts, and azodicarbonamide (ADC). The schematics of PU foams, the dynamic bond exchange of PU foams, and the structure of the carbamate exchange catalyst are illustrated in Figure 11. The process yields recycled PU with a porous structure. The continuity of the reforming process was evidenced by a consistent cell diameter. The recycled foams exhibited good mechanical properties. PU foams can thus be recycled to new foams. Designing PU foams with built-in recyclability is crucial, as it facilitates easier disassembly and reprocessing, thereby promoting a circular economy model.

## 4. Conclusions

Polyurethane (PU) foams have garnered significant interest due to their diverse properties and applications. This review primarily addresses key aspects of multifunctional and high-performance applications of PU foams. Owing to their low density and favorable mechanical properties, PU foams are predominantly utilized in flame retardancy, sound absorption, radar absorption, EMI shielding, shape memory, and biomedical applications. The increasing demand for materials that can perform multiple functions while maintaining or enhancing their core properties has made PU foams a focal point of interest for both engineers and researchers. However, to meet the demands of increasingly discerning consumers, the range of properties and applications of PU foams must be expanded, which can be achieved through innovative process technologies or the advancement of PU foam production. It is also essential to note that the production of PU foams remains highly dependent on petroleum. Considering increasingly stringent regulations, alternative raw materials derived from renewable resources for PU foam production must be studied. New process technologies and further developments in sustainability and composites are poised to significantly influence the industry in the near future. Consequently, PU foams hold significant potential for multifunctional and high-performance applications. With continued research and development, they are expected to play an increasingly important role across various industries while addressing the associated environmental challenges.

## Figures and Tables

**Figure 1 polymers-16-03182-f001:**
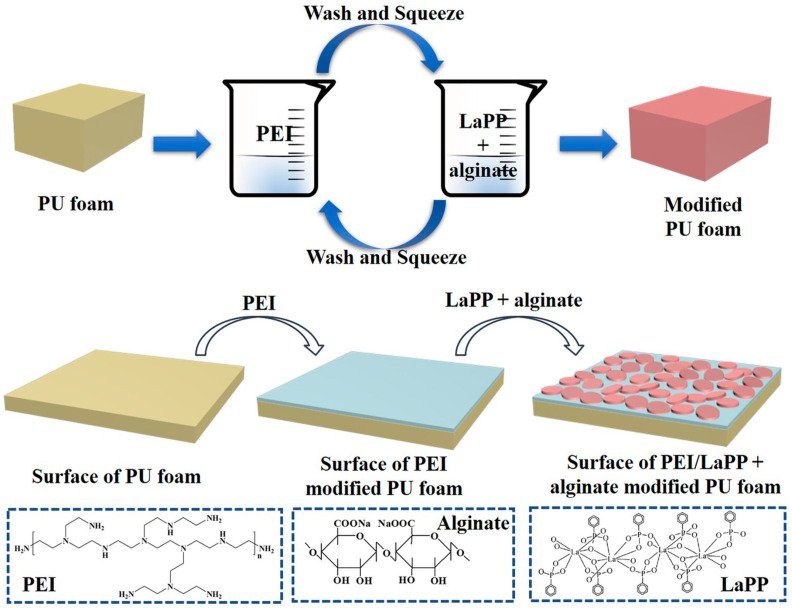
Schematic of fabrication of coatings on the PU foams [19].

**Figure 2 polymers-16-03182-f002:**
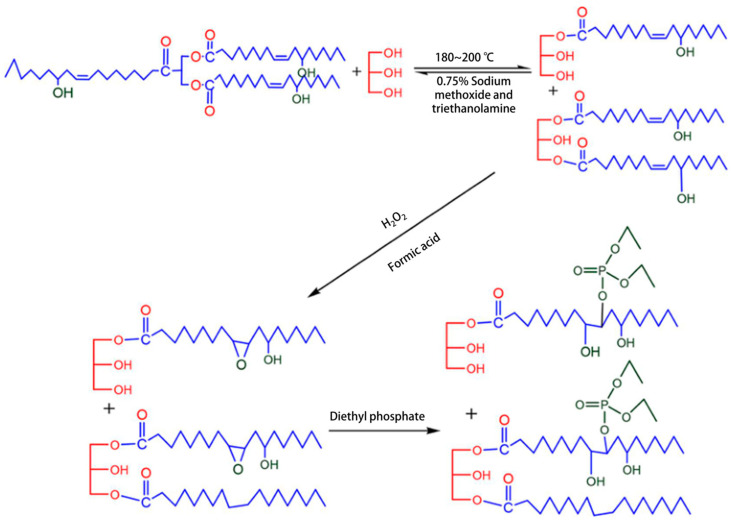
The synthesis of flame-retardant polyol [21].

**Figure 3 polymers-16-03182-f003:**
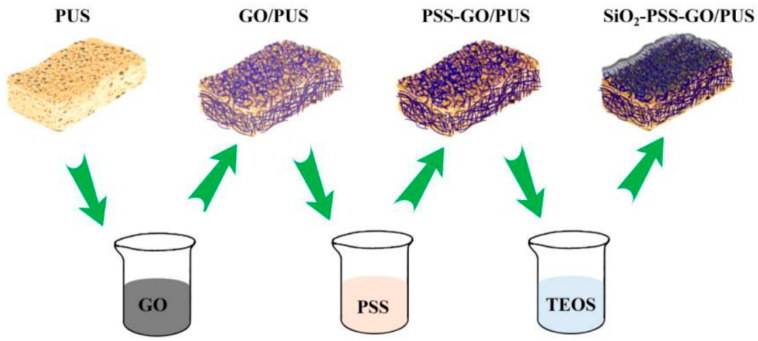
Schematic of modified PUS preparation process [29].

**Figure 4 polymers-16-03182-f004:**
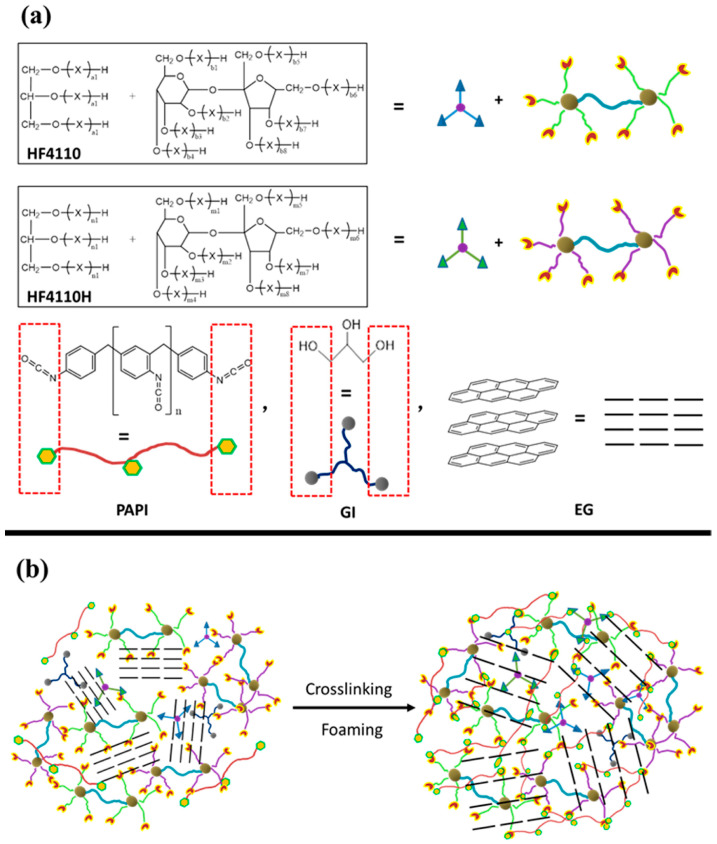
(**a**) The components of PU foams with EG, (**b**) the crosslinking structure and foaming processes of the PU/EG foams [32].

**Figure 5 polymers-16-03182-f005:**
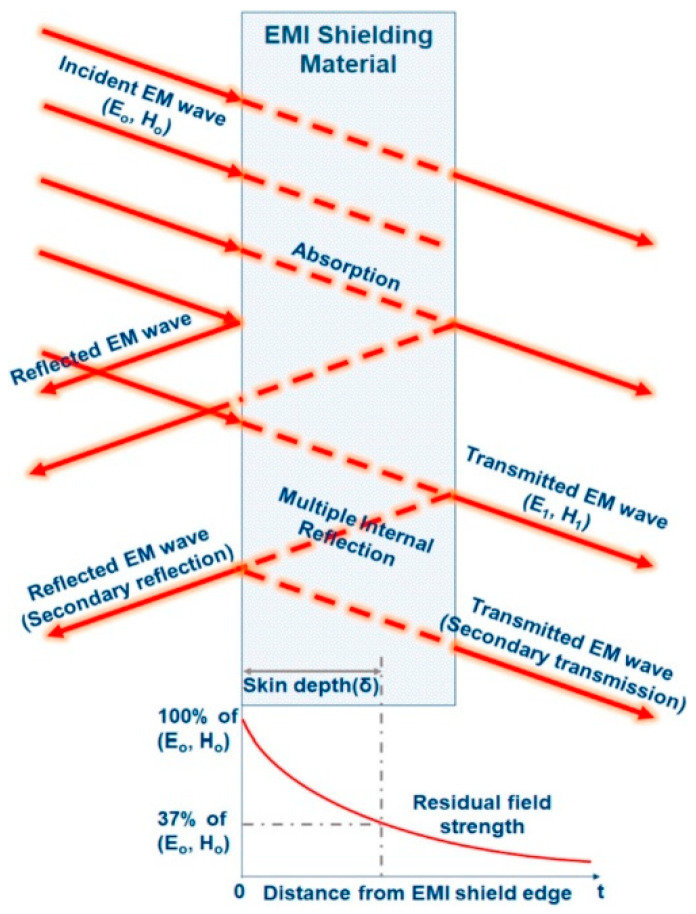
Schematic of EMI shielding mechanism [33].

**Figure 6 polymers-16-03182-f006:**
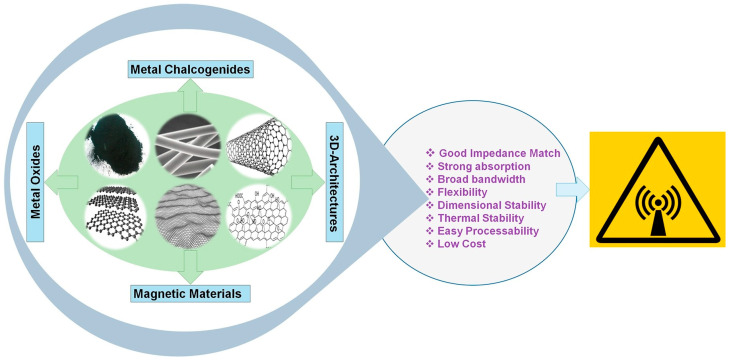
Carbon-based fillers for EMI shielding effectiveness [33].

**Figure 7 polymers-16-03182-f007:**
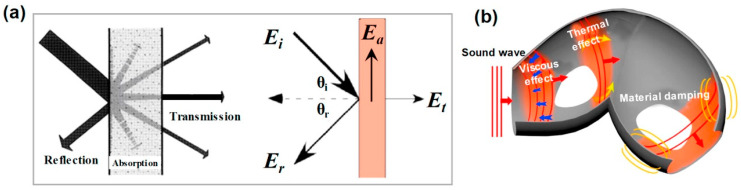
(**a**) Schematic of the sound absorption for PU foams. (**b**) Schematic of the energy consumption mechanisms of PU foams [40].

**Figure 8 polymers-16-03182-f008:**
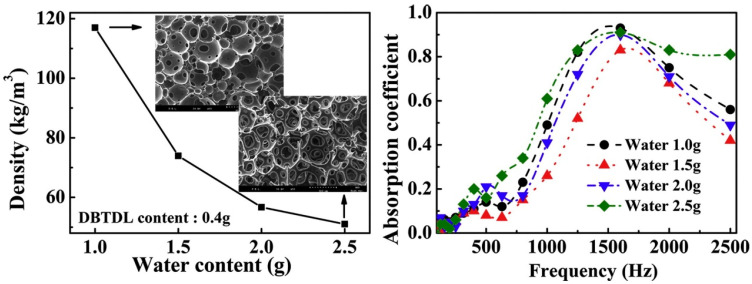
SEM images of PU foams with two gelling catalysts, and sound-damping performance of PU foams at various water contents [39].

**Figure 9 polymers-16-03182-f009:**
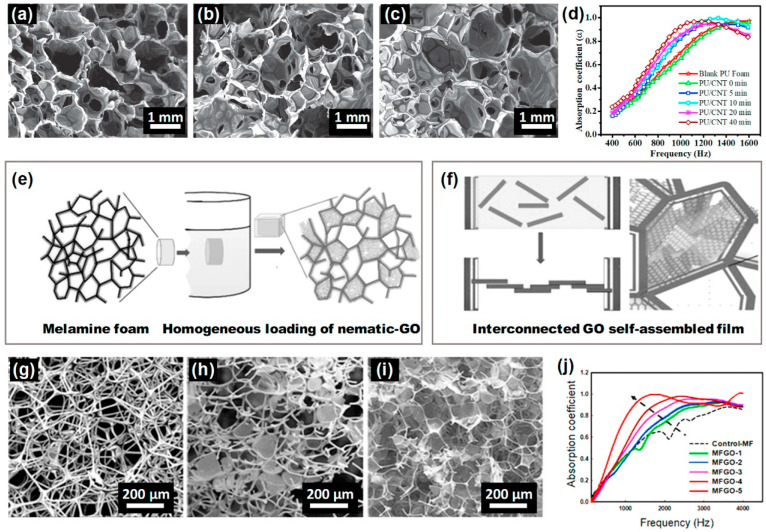
SEM images of (**a**,**b**,**g**) PU foams, (**c**) PU foams with sonication. (**d**,**e**) Sound absorption performance [43]. (**f**) Scheme of microscopic GO. (**h**,**i**) PU foams with GO. (**j**) Sound absorption performance [44].

**Figure 10 polymers-16-03182-f010:**
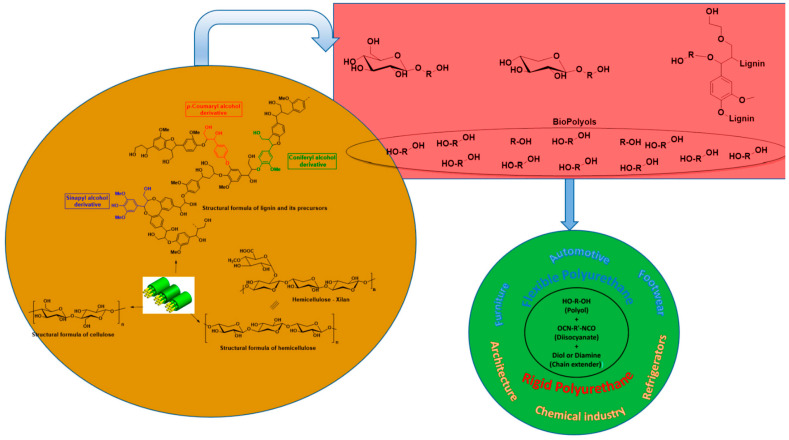
Structuring of lignocellulose biomass and structural formula of the component of PU foams [54].

**Figure 11 polymers-16-03182-f011:**
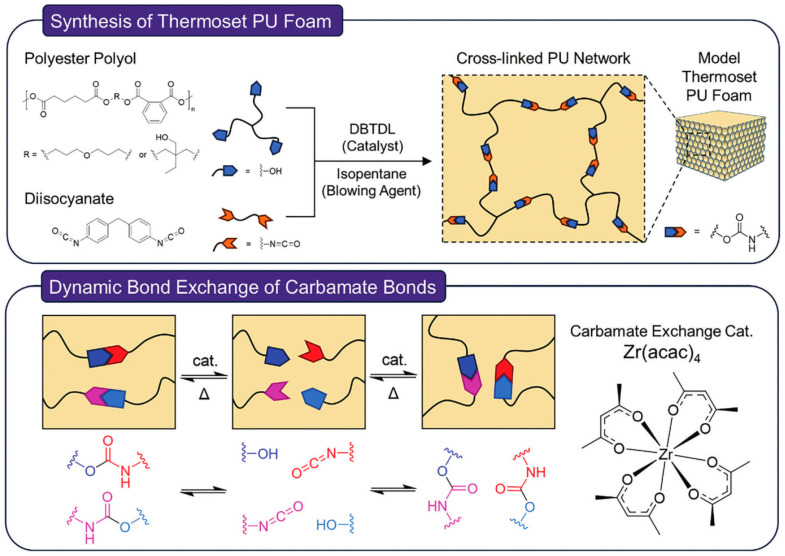
Schematics of PU foams synthesis, dynamic bond exchange in PU foams, and structure of carbamate exchange catalyst [1].

## Data Availability

No new data were created or analyzed in this study.

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
