# Peer review of "A Review of Polyurethane Foams for Multi-Functional and High-Performance Applications"

_polymers, 2024, doi:10.3390/polym16223182_

Round 1

Reviewer 1 Report

Comments and Suggestions for Authors

The author should be very focused on uploading the correct manuscript since the current version has some comments included in the manuscript from the authors or the supervisor which has to be removed. 

Comments for the author, 

1. What are the main features of polyurethane (PU) foams that make them good for high-performance and multifunctional uses?

2. What effects do the soft and hard parts of PU foams have on their ability to resist heat and force?

3. How has the process of making flame-resistant PU foams changed over the years, and what kinds of materials are used?

4. Based on your experience, how well do PU foams block electromagnetic interference (EMI)? What makes them good at this job?

5. What role do nanofillers play in making PU foams better, especially when it comes to how well they absorb sound and how strong they are?

What problems does the PU foam business have with sustainability and recycling, and what are some ideas for how to fix them?

7. What changes can be made to the chemicals in PU foams and how they are processed to make their features fit different needs?

PU foams are used in biological uses in what ways? What benefits do they offer in this field?

9. What problems does making and throwing away standard PU foam cause for the environment? How is the study working to find sustainable alternatives?

10. What are some ideas for where PU foam research and development should go in the future to meet both performance needs and environmental rules?

Author Response

Response to the comments of Reviewer 1

  1. What are the main features of polyurethane (PU) foams that make them good for high-performance and multifunctional uses?

Response:

Thank you for your comments. PU foams have low density and good dimensional stability due to their hollow cellular structure and chemical composition. PU foams have low density and good dimensional stability. PU foams have low density and good dimensional stability due to their hollow cellular structure and chemical composition. PU foams have low density and good dimensional stability. The performance of PU is primarily determined by its molecular structure and the aggregation state, which consists of soft and hard phases. The soft phases should be in proper ratio with hard phases for the optimum performance of PU. For multifunctional uses, PU foams can be tailored to specific requirements by adjusting their chemical composition conditions. This flexibility enables the creation of foams with varying densities, hardness, flame retardancy, sound absorption, radar absorption, and electromagnetic interference shielding.

  1. What effects do the soft and hard parts of PU foams have on their ability to resist heat and force?

Response:

Thank you for your comments. The soft segments provide flexibility for the PU molecular chain and present the adaptability to external forces in terms of mechanical properties, making the product flexible and stretchable. The hard parts, on the other hand, are rigid and show an increase in strength in terms of mechanical properties. There are many hydrogen bonds in the hard parts, and there is a strong electrostatic force, which promotes the aggregation of the hard segments, resulting in micro-phase separation. The higher the purity of the hard segments, the higher the degree of microphase separation, and the softening temperature of the polyurethane increases.

  1. How has the process of making flame-resistant PU foams changed over the years, and what kinds of materials are used?

Response:

Thank you for your comments. Traditional halogen-based flame retardants need to be replaced due to their toxicity and environmental impact. The surface modification process and the development of new organic flame-retardant components have become two popular methods, that is, the use of some flame-retardant components on the surface of the foam for modification, or the modification of the basic components of the foam. Commonly used materials include modified polyols containing phosphorus and nitrogen, graphene oxide, phosphoric acid, etc. To reduce the smoke generation of polyurethane foam when burned, inorganic flame retardants including magnesium hydroxide, titanium dioxide, and silicon dioxide have been explored. Expandable graphite (EG) can reduce the effective heat of combustion (EHC) by inhibiting heat release during matrix decomposition into gas products, indicating its significant role in enhancing fire resistance in the condensed phase.

  1. Based on your experience, how well do PU foams block electromagnetic interference (EMI)? What makes them good at this job?

Response:

Thank you for your comments. PU composite foams are suitable for use in the field of absorbing shielding due to their lightweight, high processability, and insensitivity to corrosive environments. The electromagnetic interference (EMI) shielding effectiveness value of PU foam composites can reach 50~60 dB (B. Shen et al. ACS Applied Materials & Interfaces, 8, 2016, 8050-8057.), far surpassing the target level of ∼20 dB required for commercial application. To enhance the conductivity of PU foams, various conductive fillers such as carbon black, graphene, and CNTs have been introduced into the PU matrix.

  1. What role do nanofillers play in making PU foams better, especially when it comes to how well they absorb sound and how strong they are?

Response:

Thank you for your comments. The addition of nanofillers can increase the number of open structures on the foam wall, or forms a complex channel structure, which is conducive to the improvement of the sound absorption performance of the foams (M. Park et al. Journal of Nanoscience and Nanotechnology, 19, 2019, 3558-3563.). The nanofiller has excellent mechanical properties, and after being added to the PU foams, it can play a supporting role in the bubble skeleton and make the size of the bubble become small and uniform as a nucleating agent, so that the mechanical properties of the foam can be improved.

  1. What problems does the PU foam business have with sustainability and recycling, and what are some ideas for how to fix them?

Response:

Thank you for your comments. Most polyols used to produce PU foams are made from raw materials derived from petroleum, and the management of increasing PU waste presents a significant challenge. The development of sustainable PU foams focuses on reducing environmental impact by using renewable resources, enhancing recyclability. Renewable polyols sourced from biomass residues, vegetable, or industrial by-products need to be developed, and chemical recycling methods that convert waste PU foams back into their constituent monomers or other valuable chemicals deserve more attention (A. M. Thakker et al. Environmental Science and Pollution Research, 30, 2023, 101989-102009.).

  1. What changes can be made to the chemicals in PU foams and how they are processed to make their features fit different needs?

Response:

Thank you for your comments. PU foams can be tailored to specific requirements by adjusting their chemical composition conditions (J. Zhang et al. Journal of Applied Polymer Science, 138, 2021, 50583.). For rigid foams, it is usually necessary to reduce the molecular weight of the soft segment, select a rigid isocyanate type, and add a reinforcing filler. For flexible foams, polyols with higher molecular weights are generally used. For a product with high resilience, it is necessary to increase the degree of cross-linking appropriately. The addition of flame retardants and conductive fillers can make the PU foams have a certain flame-retardant effect or conductive ability.

  1. PU foams are used in biological uses in what ways? What benefits do they offer in this field?

Response:

Thank you for your comments. There are many applications of PU foams in biology, including drug delivery, tissue engineering, and long-lasting biomedical devices (C. Ribeiro et al. Colloids and Surfaces B: Biointerfaces, 136, 2015, 46-55.). The injectable delivery techniques can be used to prepare PU foam scaffolds. PU foams have emerged as significant polymers for biomedical applications because of their good properties, including biocompatibility, and controllable chemical and physical characteristics. They also have good mechanical properties as well as processing properties. Modification of PU foams broadened their applicability in biomedical contexts, leading to their use in drug delivery, scaffolds, and stents for tissue, bio-fluid absorption, or biocatalytic air filters (Y. C. Shin et al, Journal of Biomaterials science, Polymer Edition, 29, 2018, 762-774.).

  1. What problems does making and throwing away standard PU foam cause for the environment? How is the study working to find sustainable alternatives?

Response:

Thank you for your comments. Other raw materials, such as polyols derived from petrochemical products, are used in the manufacture of conventional polyurethane foams, increasing the pressure on the exploitation of non-renewable resources such as petroleum. At the same time, after chemical polymerization and cross-linking, these foams are usually difficult to degrade and cannot be recycled, resulting in a waste of resources and environmental pollution. The development of sustainable PU foams focuses on reducing environmental impact using renewable resources, enhancing recyclability, and incorporating multifunctional properties. The foundation of advancing sustainable PU foams lies in transitioning from fossil-derived raw materials to bio-based substitutes. Bio-based substitutes have a lower carbon footprint and can be recycled or sourced from non-polluting sources (P. Cinelli et al, European Polymer Journal, 49, 2013, 1174-1184.).

  1. What are some ideas for where PU foam research and development should go in the future to meet both performance needs and environmental rules?

Response:

Thank you for your comments. To obtain both performance and environmental benefits, alternative raw materials derived from renewable resources for PU foam production must be considered. Given the current performance problems of raw materials derived from biomass, it is necessary to continue to explore the modification of raw materials and appropriate processing methods (H. J. Kim et al, Scientific Reports, 14, 2024, 13490.). At the same time, there is an urgent need to solve the problems of bio-based raw materials in terms of yield and processing cost to meet the needs of large-scale production. When designing recyclable foam products, it is necessary to pay attention to the balance between recycling performance and product mechanical properties or heat resistance, so that they can be applied in more fields, accelerating the promotion of environmentally friendly materials (A. Delavardea et al. Progress in Polymer Science, 151, 2024, 101805.).

Reviewer 2 Report

Comments and Suggestions for Authors

The manuscript provides a review of the applications and development of polyurethane (PU) foams. While no significant flaws are identified, some reviews lack depth and appear superficial. To enhance the manuscript's suitability for publication, I recommend the authors address the following issues:

Major Issues:

  1. The manuscript does not include information on funding sources or author contributions.
  2. The authors claim that the paper examines the challenges faced by the PU foam industry, but it lacks sufficient detail in this regard. For instance, while the authors mention the industry's focus on transitioning from fossil-derived raw materials to bio-based substitutes like lignocellulose, they do not address the specific obstacles to this transition. What challenges need to be overcome? What progress has been made? What research methods have contributed to this advancement?

Minor Issues:

  1. The sentence on lines 75-76 may be missing a word: "This of PU foams allow for their adaptation to meet specific needs by modifying their..."
  2. On lines 150-151: "the higher this ratio, the greater the hazardousness of the material when burned." Please clarify which ratio is being referred to.
  3. Please correct the first instance of Fig. 9 to Fig. 8.

Author Response

Response to the comments of Reviewer 2

The manuscript provides a review of the applications and development of polyurethane (PU) foams. While no significant flaws are identified, some reviews lack depth and appear superficial. To enhance the manuscript's suitability for publication, I recommend the authors address the following issues:

Major Issues:

The manuscript does not include information on funding sources or author contributions.

Response:

Thank you for your suggestion. We have added relevant information in the manuscript.

Funding sources:

This work received no external funding.

Author Contributions:

Conceptualization, methodology, project administration, funding acquisition, Yongjun Chen, Jiang Jiang, Yuanfang Luo. Writing—original draft, investigation, visualization analysis, Huanhuan Dong, Shujing Li, Zhixin Jia; Sheng Ji. Writing—review and editing, Huanhuan Dong, Shujing Li. Resources, project administration, Yuanfang Luo. All authors have read and agreed to the published version of the manuscript.

The authors claim that the paper examines the challenges faced by the PU foam industry, but it lacks sufficient detail in this regard. For instance, while the authors mention the industry's focus on transitioning from fossil-derived raw materials to bio-based substitutes like lignocellulose, they do not address the specific obstacles to this transition. What challenges need to be overcome? What progress has been made? What research methods have contributed to this advancement?

Response:

Thank you for your suggestion. There are several challenges met by the development of bio-based substitutes. PU foam products obtained with biomass raw materials as a component may have the disadvantage of insufficient performance. Some bio-based alternatives also face low yields and higher processing costs. The process of synthesizing some bio-based foams is complex, which limits its large-scale application.

Some works have been carried out to make bio-base PU achieve performance that rivals or even exceeds that of traditional petrochemical-based PU. Researchers (B. Z. Zhao et al. Chemical Engineering Journal, 494, 2024, 152941.) have prepared cardanol-based CPU foams with good self-healing ability, and the vertical and parallel compressive strengths reach up to 0.27 MPa and 0.32 MPa, respectively, when the addition of CBPs was 80 %. The issue of cost has also been taken into consideration. Plant-derived polyols have been drawing attention due to the high availability, low cost, and renewable nature of vegetable oils. Ahir et al. (M. Ahir et al. Polymers, 16, 2024, 1584.) synthesize hemp bio-polyols by a ring-opening reaction, followed by the production of RPUF in a single step. The effect of melamine as a flame retardant in composite foam was also examined, which shows the highest compression strength of 447 KPa. Other methods, such as modification, partial substitution (J. Zhou et al. Materials Today Communications, 41, 2024, 110438.), combination of two bio-based substitutes (S. Silvano et al. International Journal of Biological Macromolecules, 278, 2024, 135282.) and addition of functionalized fillers (S. R. Acharya et al. Materials Today Communications, 41, 2024, 110278.), have been used to expand the application field of bio-based PU.

Minor Issues:

The sentence on lines 75-76 may be missing a word: "This of PU foams allow for their adaptation to meet specific needs by modifying their..."

Response:

Thank you for your suggestion. We correct the sentence on lines 75-76. The original sentence “This of PU foams allow for their adaptation to meet specific needs by modifying their...” is revised to “This characteristic of PU foams allows for their adaptation...”

On lines 150-151: "the higher this ratio, the greater the hazardousness of the material when burned." Please clarify which ratio is being referred to.

Response:

Thank you for your suggestion. The ratio there means the effective heat of combustion (EHC). We correct the sentence on lines 150-151. The original sentence “the higher this ratio, the greater the hazardousness of the material when burned.” is revised to “The higher the EHC, the greater the hazardousness of material when burned.”

Please correct the first instance of Fig. 9 to Fig. 8.

Response:

Thank you for your suggestion. We have corrected the first instance of Fig. 9 to Fig. 8.

Round 2

Reviewer 1 Report

Comments and Suggestions for Authors

The article is revised and sounds good. The paper can be accepted for publication. 

Reviewer 2 Report

Comments and Suggestions for Authors

The authors have addressed all my conerns and no further revision is needed.